# Differentiable Optimization of Similarity Scores Between Models and Brains

Nathan Cloos[1]  Moufan Li[2]  Markus Siegel[3]  Scott L. Brincat[1]
Earl K. Miller[1]  Guangyu Robert Yang[1]  Christopher J. Cueva[1]

[1]MIT  [2]NYU  [3]HIH Tübingen
nacloos@mit.edu, ccueva@gmail.com

## Abstract

How do we know if two systems – biological or artificial – process information in a similar way? Similarity measures such as linear regression, Centered Kernel Alignment (CKA), Normalized Bures Similarity (NBS), and angular Procrustes distance, are often used to quantify this similarity. However, it is currently unclear what drives high similarity scores and even what constitutes a "good" score. Here, we introduce a novel tool to investigate these questions by differentiating through similarity measures to directly maximize the score. Surprisingly, we find that high similarity scores do not guarantee encoding task-relevant information in a manner consistent with neural data; and this is particularly acute for CKA and even some variations of cross-validated and regularized linear regression. We find no consistent threshold for a good similarity score – it depends on both the measure and the dataset. In addition, synthetic datasets optimized to maximize similarity scores initially learn the highest variance principal component of the target dataset, but some methods like angular Procrustes capture lower variance dimensions much earlier than methods like CKA. To shed light on this, we mathematically derive the sensitivity of CKA, angular Procrustes, and NBS to the variance of principal component dimensions, and explain the emphasis CKA places on high variance components. Finally, by jointly optimizing multiple similarity measures, we characterize their allowable ranges and reveal that some similarity measures are more constraining than others. While current measures offer a seemingly straightforward way to quantify the similarity between neural systems, our work underscores the need for careful interpretation. We hope the tools we developed will be used by practitioners to better understand current and future similarity measures.

Project page      Code

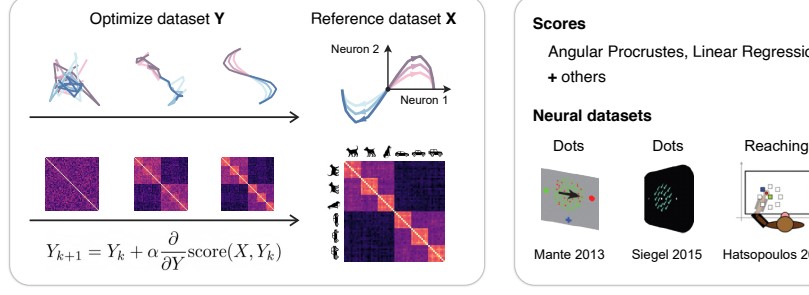

Figure 1: *(a)* **To better understand the properties of similarity measures we optimize synthetic datasets to become more similar to a reference dataset**, for example, neural recordings. *(b)* We analyzed similarity scores between artificial datasets and electrode recordings from five experiments on nonhuman primates spanning a diverse range of behaviors and brain regions.

## 1 INTRODUCTION

Similarity measures have become a cornerstone in evaluating representational alignment across different models (Kornblith et al., 2019), different biological systems (Kriegeskorte et al., 2008b), and across both artificial and biological systems. Researchers have employed diverse methods to compare model representations with, for example, brain activity, aiming to identify models that exhibit brain-like representations (Yamins et al., 2014; Sussillo et al., 2015; Schrimpf et al., 2018; Nayebi et al., 2018). However, while these measures are actively used and provide an efficient way to compare structure across complex systems, it is not clear that they adequately represent the computational properties of interest, and there is a need to better understand their limitations. Whenever we choose a similarity measure, we are making a commitment to what we care about in the two systems we are comparing. Therefore, understanding what drives these similarity scores is crucial for understanding this commitment. The field lacks clear guidelines for interpreting similarity scores in a given experimental context.

In this work we study several popular methods that have been proposed to quantify the similarity between models and neural data, in particular, linear regression (Yamins et al., 2014; Schrimpf et al., 2018), Centered Kernel Alignment (CKA) (Kornblith et al., 2019), angular Procrustes distance (Williams et al., 2021; Ding et al., 2021), and Normalized Bures Similarity (NBS) (Tang et al., 2020). See appendix C.1 for a brief overview. We analyzed neural data from studies on nonhuman primates (Figure 1) and compared the neural responses to task-optimized recurrent neural networks (RNNs), or synthetic datasets, with different similarity scores. In order to study what drives high similarity scores we directly optimize the synthetic datasets to maximize their similarity to the neural datasets as assessed by different similarity measures.

**Disagreement between similarity measures.** An example application of similarity measures is to quantify how brain-like models are. However, when we compare task-optimized RNNs to two neural datasets, as shown in Figure 2, we find that different similarity measures do not agree about which models are more similar to the data, or even about whether the models are more or less similar than two baseline scores that compare modified versions of the neural data to the original neural data. Are most of the models generally performing well or not, i.e. achieving model-data similarities above one or both of these baseline scores? Different measures give different answers. These similarity measures lack consistency and do not present a consensus interpretation. This is not an irrelevant exercise as all of these similarity measures have been used in the literature to compare models to neural data.

Our main **contributions** and findings can be summarized as follows:

1. We show that a good value for a similarity score varies depending on the similarity measure and the dataset (Figure 3). Furthermore, we demonstrate that a high similarity score does not *guarantee* that models encode task-relevant information in a manner consistent with neural data. This is particularly relevant for the many studies that rely on these similarity measures to quantitatively characterize the degree of alignment between models and brains.

2. We identify what drives high similarity scores by differentiating through the similarity measures to directly maximize the score. We discover that different similarity measures differentially prioritize learning principal components of the data. For example, CKA, as opposed to angular Procrustes, may indicate a high score when many of the lower variance components are not captured, even when these dimensions carry crucial task-related information (Figures 4 and 5).

3. Through theoretical derivation we show the sensitivity of CKA, angular Procrustes, and Normalized Bures Similarity to the variance of principal component dimensions, and explain the dependence CKA shows to high variance components (section 4.3 and Figure 6)

4. We characterize the allowable range of scores between two different similarity measures by jointly optimizing their scores. Surprisingly, we reveal that a high angular Procrustes similarity implies a high CKA score, but not the converse (for the dataset shown in Figure 7).

5. Comparing similarity scores across studies is challenging, primarily due to variability in naming and implementation conventions. As part of our contribution to the research community we have created, and are continuing to develop, a Python package that benchmarks and standardizes similarity measures. Currently there are approximately 100 different similarity measures from 14 packages. **Similarity package:** https://github.com/nacloos/similarity-repository

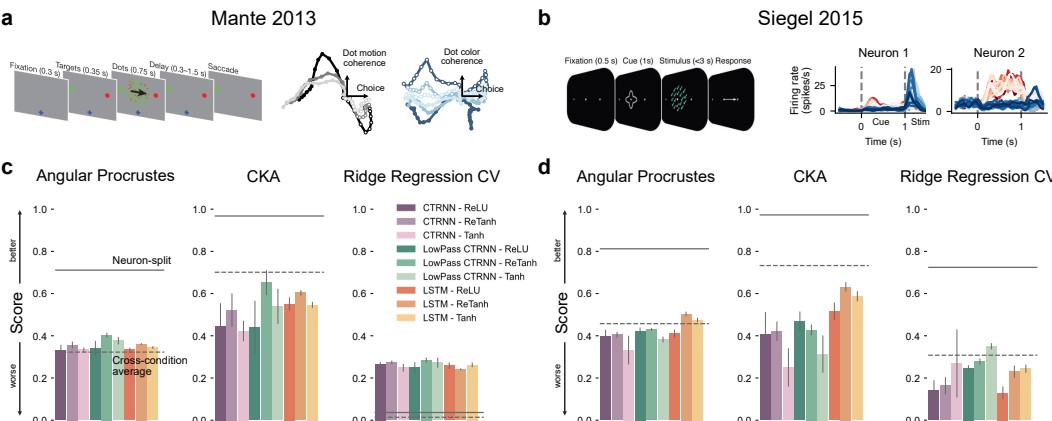

Figure 2: **Different similarity measures do not agree on the relative rankings when comparing models to neural datasets.** One example application of similarity measures is to evaluate the similarity of task-optimized recurrent neural networks to neural datasets. We consider two neural datasets from *(a)* prefrontal cortex (PFC) (Mante et al., 2013) and *(b)* Frontal Eye Field (FEF) (Siegel et al., 2015) in monkeys performing an experimental task that required the animal to attend to either color or motion information while ignoring the non-cued feature of the stimuli. *(c, d)* RNNs with three different architectures, CTRNN, LowPassCTRNN, LSTM and three different nonlinearities, ReLU, ReTanh, Tanh are compared to neural datasets (see appendix B for details).

## 2 RELATED WORK

**Reviews.** Recent reviews by Sucholutsky et al. (2024) and Klabunde et al. (2023) provide comprehensive overviews of representational similarity measures and their theoretical properties. While these reviews highlight the diversity of available metrics, they offer limited practical guidance on interpreting similarity scores.

Our work addresses this gap by proposing a general framework for evaluating similarity measures. By directly optimizing synthetic datasets to maximize their similarity to, for example, neural recordings, we can systematically investigate how different metrics prioritize various aspects of the data, such as specific principal components or task-relevant information. Similarity measures are often characterized by the invariance properties between representations with maximum similarity. In contrast, we focus on what drives intermediate similarity scores and how to interpret them, since we are often dealing with intermediate levels of similarity in practice when comparing models to brain data.

**Mathematical properties.** This work builds upon several important theoretical and empirical contributions. Kornblith et al. (2019) discussed the invariance properties of similarity measures and their implications for comparing neural representations. Williams et al. (2021) advocated for similarity measures that satisfy the axioms of a metric distance. Harvey et al. (2023) established a duality between Normalized Bures Similarity (NBS) and Procrustes distance, shedding light on their mathematical relationship to CKA. We leverage these theoretical insights to provide a deeper interpretation of our empirical findings.

**Comparison with functional measures.** Our approach shares similarities with the work of Ding et al. (2021), who evaluated metrics based on their correlation with functional behavioral measures. They analyzed a collection of trained models and observed how variations in the models, such as the removal of principal components, impacted both the similarity scores and the performance on task-specific probes. They identified that CKA exhibited sensitivity primarily to the top principal components, while orthogonal Procrustes demonstrated more robust performance.

However, our approach differs significantly in how we construct a diverse set of datasets with varying behavioral characteristics. Instead of relying on pre-trained models, we begin with unstructured noise and directly optimize it to maximize similarity to neural recordings. This allows us to address questions such as whether high similarity scores can be achieved without encoding task-relevant variables. Our optimization-based approach reveals how different similarity measures guide the emergence of task-relevant information within the synthetic datasets, offering a dynamic perspective

on their properties. Additionally, by leveraging our optimization-based perspective for multiple similarity measures we can find the allowable ranges of scores, and identify dependencies between similarity measures.

In contrast to the majority of previous work, our method is model-agnostic, focusing on the properties of similarity measures themselves rather than specific model architectures. We demonstrate the generalizability of our findings by applying our framework to multiple neural datasets, revealing consistent patterns in how different metrics prioritize data features. Interestingly, we find that the optimization dynamics observed with neural data are closely predicted when using Gaussian datasets with matched variance distributions. This suggests that our insights extend beyond the specifics of individual neural datasets and reflect fundamental properties of the similarity measures themselves.

## 3 METHOD

### 3.1 MEASURING SIMILARITY

To evaluate the similarity of representations between two systems, we extract feature representations such as activity in a brain area or model layer in response to some sample stimuli. Our objective is to quantify the alignment between these representations using a similarity score. Assume two datasets $X$ and $Y$ represent these features with dimensions (sample, feature) and mean-centered columns. Datasets with temporal dynamics are reshaped from (time, sample, feature) to (time*sample, feature). We define a scoring function $\text{score}(X, Y)$ as a measure that increases with similarity, achieving a maximum of 1 when $X = Y$.

We consider the following similarity measures (see appendix C.1 for details):

- **Centered Kernel Alignment (CKA)** measures the correlation between the kernels of two datasets $X$ and $Y$ (Kornblith et al., 2019). We consider here linear CKA where the kernels are $XX^T$ and $YY^T$.

$$\text{CKA}(X, Y) = \frac{\langle \text{vec}(XX^T), \text{vec}(YY^T) \rangle}{\|XX^T\|_F \|YY^Y\|_F}$$

- **Angular CKA** is a variant of CKA that satisfies the axioms of a distance metric (Williams et al., 2021; Lange et al., 2022). It is defined as the arccosine of CKA. We additionally renormalize it to have a value between 0 and 1, where 1 is perfect similarity.

- **Angular Procrustes** finds the optimal orthogonal linear alignment between $X$ and $Y$ to maximize their correlation (Williams et al., 2021; Ding et al., 2021).

$$\min_{Q \in \mathcal{O}} \arccos \frac{\langle X, QY \rangle}{\|X\|_F \|Y\|_F}$$

where $\mathcal{O}$ is the group of orthogonal linear transformations. Williams et al. (2021) proposed taking the arccosine to satisfy the axioms of a distance metric. Here we rescale the angular Procrustes distance to obtain a scoring measure between 0 and 1, where 1 is perfect similarity.

- **Normalized Bures Similarity (NBS)** (Tang et al., 2020) is the cosine of the angular Procrustes distance (Harvey et al., 2023). NBS is also closely related to CKA and mainly differs by a choice of matrix norm (see details in appendix C.1).

$$\text{NBS}(X, Y) = \frac{\|X^T Y\|_*}{\sqrt{\|XX^T\|_* \|YY^T\|_*}}$$

- **Ridge Regression** finds the best linear mapping $B$ that predicts a reference dataset $X$ from a dataset $Y$ (Yamins et al., 2014; Schrimpf et al., 2018). We measure the goodness of fit using $R^2$ and use ridge regularization as well as 5-fold cross-validation that tests generalization across different experimental conditions. We use $\lambda = 100$ by default and show results with varying $\lambda$ in appendix A.

$$B^* = \arg\min_B \|X - YB\|_F^2 + \lambda \|B\|_F$$

$$R_{LR}^2 = 1 - \frac{\|X - YB^*\|_F^2}{\|X\|_F^2}$$

- **Linear Regression.** Same as Ridge Regression but for $\lambda = 0$ (unregularized).

## 3.2 Optimizing similarity scores

To better characterize similarity measures we optimize synthetic datasets $Y$ to become more similar to a reference dataset $X$. We initialize the synthetic dataset $Y$ by randomly sampling from a standard Gaussian distribution with the same shape as $X$. We use Adam (Kingma & Ba, 2017) to optimize $Y$ to maximize the similarity score with $X$, leveraging the differentiability of the similarity measures, and stop the optimization when the score reaches a fixed threshold near 1. Note that some similarity measures have parameters to optimize to compute the similarity score. Our method can be applied in such cases too, as long as the similarly score is differentiable with respect to the input datasets. For example, in the case of linear regression, we directly differentiate PyTorch's lstsq function.

As we optimize $Y$ towards greater similarity with $X$, we simultaneously evaluate how well task-relevant variables can be linearly decoded from the evolving synthetic data. This decoding analysis employs logistic regression with stratified 5-fold cross-validation. For datasets with temporal dynamics, we fit a separate decoder for each time step and report the average accuracy across time.

We also evaluate how each principal component (PC) of the reference dataset $X$ is captured as $Y$ is optimized for similarity. Specifically, for each PC $v_i$ of $X$, we use linear regression to find the optimal linear combination of columns of $Y$ that best predicts $Xv_i$. The goodness of fit is then quantified using the $R^2$ coefficient:

$$R^2_{PC_i} := 1 - \frac{\|(X - \hat{X})v_i\|^2}{\|Xv_i\|^2}$$

where $\hat{X} = Y\hat{B}$ and $\hat{B} = \text{argmin}_B \|X - YB\|_F^2$.

## 4 Results

### 4.1 What constitutes a good similarity score depends on the similarity measure and the dataset

**What is a good value for a similarity score?** Our approach to address this question is to examine, across five neural datasets, the similarity score required for a synthetic dataset to encode task relevant information to the same degree as the neural data. More specifically, the dots along the x-axis in Figure 3 indicate the score when a decoder trained on the synthetic data can extract 90% of the full task relevant information present in the neural data. Consider the Mante 2013 dataset, and notice that the "good" scores for the six similarity measures are quite different, ranging from less than 0.5 to almost 1. So even for the same dataset, the similarity score required to encode task relevant information to a similar extent as the neural data, can be very different across similarity measures.

Now if we look across the five neural datasets, at a single similarity measure, we can see that what constitutes a good score varies depending not only on the similarity measure but also on the dataset. An angular Procrustes score above 0.5 may constitute a good score for the Mante 2013 dataset but a score above 0.8 is required for the Siegel 2015 dataset (see also Supplementary Figure S1).

**High similarity scores do not guarantee encoding of task relevant variables.** A crucial point to make with Figure 3 is that high similarity scores near the maximum value of 1, particularly for CKA and unregularized linear regression without cross-validation, do not guarantee that models encode task-relevant information in a manner consistent with neural data, i.e. the CKA and linear regression curves in the Siegel 2015 dataset do not approach the horizontal line showing the decode accuracy for the neural data. There may be important features in a dataset that are not captured by a model even when the model-data similarity score is high.

For linear regression, even cross-validated and regularized, a high similarity score does not necessarily mean that the task relevant information is encoded in a similar manner to the neural data (Supplementary Figure S2).

Surprisingly, in Figure 3 some optimized datasets encode task relevant information better than the original neural data (colored lines showing decode accuracy are above the horizontal dashed line). This might suggest that the optimized datasets are denoised versions of the original data. We tested this hypothesis by removing low-variance principal components from the neural dataset, but this did not change the results. This suggests it is probably a more complex form of denoising and would require further investigation (Supplementary Figure S3).

Figure 3: **What constitutes a good score varies depending on the similarity measure and the dataset.** Decode accuracy for experimental variables versus similarity scores. The experimental variables are color vs motion contexts (binary variable) for Mante 2013 and Siegel 2015, reaching direction (total of 8 directions) for Hatsopoulos 2007, object categories (total of 8 categories) for MajajHong 2015, and texture vs noise categories (binary variable) for FreemanZiemba 2013. Horizontal dashed lines show the decode accuracy from the neural data (upper line) and chance level (lower line). Colored dots above the x-axis indicate the similarity scores when the decode accuracy reaches 90% midway between chance level and the decode accuracy from the reference neural dataset.

## 4.2    Optimization dynamics of similarity scores

### 4.2.1    Low-dimensional synthetic datasets

**Question.** How much of the neural data must be captured by a synthetic dataset or model before the decode accuracy reaches the level seen in the neural dataset itself? One perspective on this question is to decode the task variables from neural data after projecting onto principal components 1 through N, where principal component 1 captures the most variance. As we might expect, in order to capture all the information about the task variables, at least several principal components must be included in the decode (Figures S1c and S1d show an example of this analysis for two of the neural datasets). This motivates the following hypothesis.

**Hypothesis.** Perhaps the reason that some variations of linear regression and CKA similarity scores can be so high while the synthetic data fails to encode task variables is because these similarity measures preferentially rely on the top few principal components. In other words, if these measures only encode information from the top principal components then this may not be sufficient to encode all the task variables. We explore this hypothesis in the following set of analyses with a synthetic dataset based on the neural recordings from Mante et al. 2013.

**Figure 4a** shows the reference dataset (compare to Figure 2a). We can think of this reference dataset as a low-dimensional neural trajectory summarizing the population activity of many neurons, or alternatively, as the firing rates of two neurons over time (shown here encoding the two task variables of choice and dot motion coherence), recorded during six different experimental conditions, with the color in Figure 4a denoting the condition. **Figure 4b** shows the transformation of an initially random Gaussian noise dataset as it is optimized to maximize either the angular Procrustes or CKA similarity score with respect to the reference dataset. The score increases from an initial value near 0 to a maximum near 1 as optimization progresses, with the insets at the top of the figure showing the optimized noise dataset at various points during this procedure. The yellow curve shows how well the optimized dataset captures the first principal component of the reference dataset, as quantified by $R^2$, throughout optimization (see section 3.2 for details). Notice that the second principal component, shown in purple, is only captured at a *much higher* optimization score for CKA versus angular Procrustes. **Figure 4c** shows the same results when a synthetic Gaussian noise dataset is optimized towards the reference dataset using either linear regression similarity or angular CKA similarity (Williams et al., 2021).

**Dependence on the variance distribution.** The optimization dynamics not only depend on the similarity measure but also on the variance distribution of the reference dataset. Figure 4d shows four reference datasets with the same variance along the first principal component but decreasing variance along the second. If both dimensions have approximately equal variance then angular Procrustes, CKA, and linear regression will learn both dimensions similarly during optimization as shown by the white curve in Figure 4e. The curves are colored to indicate the fraction of variance the second principal component has relative to the first, so 1 indicates both principal components have the same

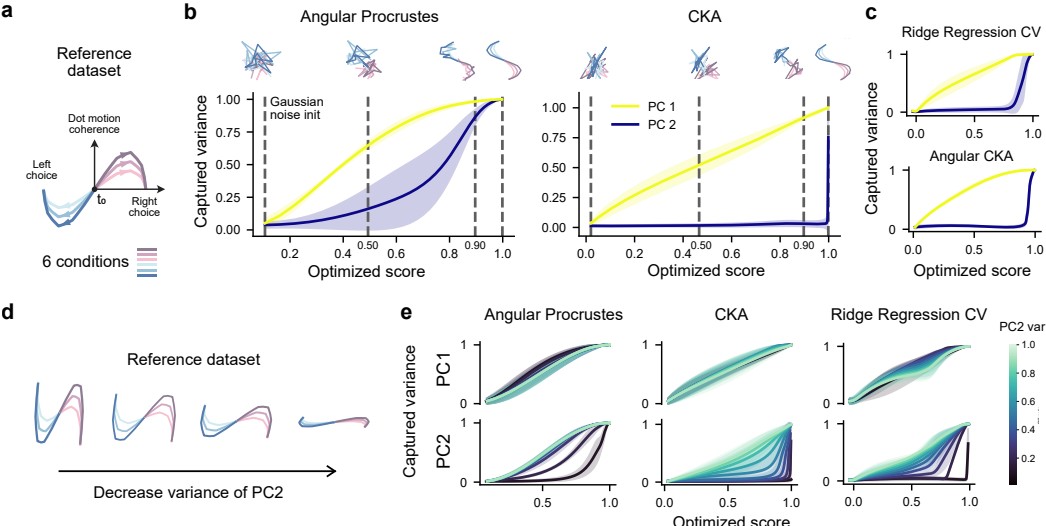

Figure 4: **Different similarity measures differentially prioritize learning principal components of the data.** *(a)* Reference dataset used as a target during optimization. *(b, c)* Initial Gaussian random noise data is updated to maximize similarity with the reference dataset, as quantified by one of the similarity measures. The transformation of the random noise dataset is shown at the top of panel *b*. The first principal component of the reference dataset is increasingly well captured by the optimized data as the similarity scores increase (yellow curves). **The second, lower variance, component is also learned when maximizing the angular Procrustes similarity but is only captured at high similarity scores when maximizing linear regression, CKA, and angular CKA similarity.** *(d)* Four reference datasets with decreasing variance along the second principal component. *(e)* Similarity measures capture both principal components when their variance is approximately equal. However, when the variance differs, CKA and linear regression preferentially neglect the low variance component (curves colored according to asymmetry of variance distribution).

variance. As the asymmetry between the principal components grows, the optimization to maximize angular Procrustes similarity still effectively learns the lower variance principal component (first column, second row), while CKA and linear regression do not capture the second principal component of the reference dataset until much later during optimization.

### 4.2.2 NEURAL DATASETS

The optimization dynamics revealed in Figure 4a for a two-dimensional dataset also holds on real neural data (B.1). We now consider several neural datasets as the reference dataset for the optimization procedure. Figure 5a shows the optimization dynamics when a random noise Gaussian dataset is optimized towards the Siegel et al. 2015 dataset by maximizing angular Procrustes similarity (top) or CKA similarity (bottom). Each curve shows how the optimized dataset captures a single principal component of the reference dataset during the course of optimization; the yellow curve shows the highest variance component. Similar to the results in Figure 4, optimizing for angular Procrustes similarity, as opposed to CKA, captures more of the lower variance components in the data for a given similarity score.

A convenient way of summarizing these curves is to note the similarity score required to capture a given principal component above some threshold. The PC threshold is defined here (and shown in Figure 5a for PC 1) as the centerpoint between the maximum and initial $R^2$ value for a given principal component. Figure 5b shows the score required to reach the principal component threshold for the different principal components in the Mante et al. 2013 dataset. Figures 5c and 5d show the same curves when the reference datasets are now the electrode recordings from Siegel et al. 2015 and Majaj et al. 2015. Figures 5b, 5c, and 5d additionally show that the metric version of CKA (angular CKA) increases CKA's sensitivity to lower variance components.

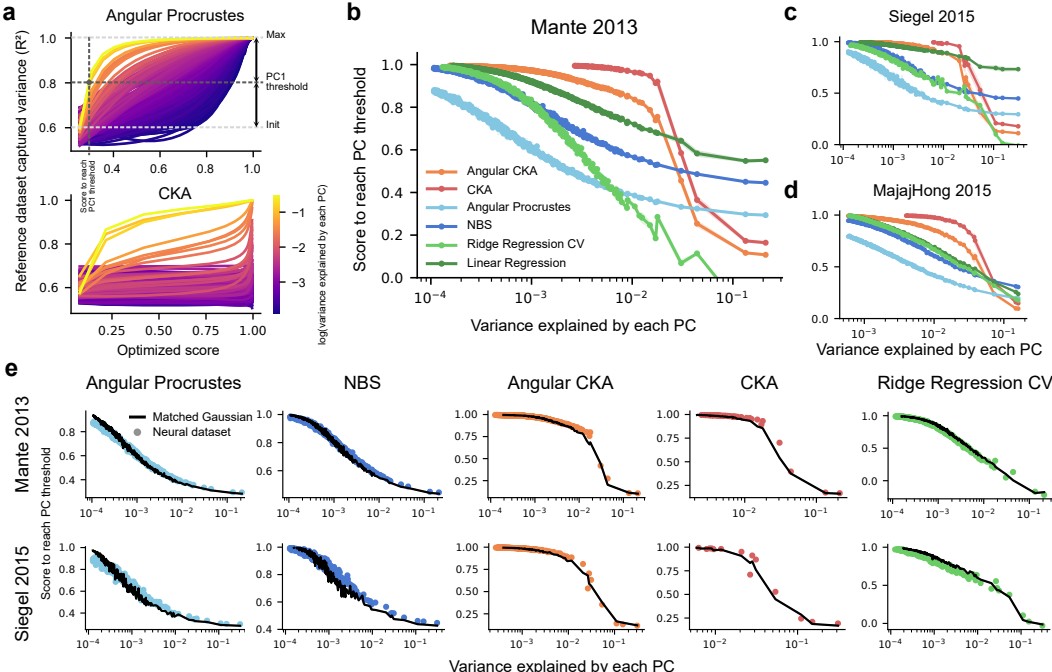

Figure 5: *(a)* **A randomly initialized synthetic dataset is updated to maximize the similarity with a neural dataset**, taken here to be the FEF dataset from (Siegel et al., 2015). The principal components (PCs) of this reference dataset are captured by the optimized dataset at different similarity scores, which in subsequent figures we call the score to reach the PC threshold. *(b)* The score to reach the PC threshold for the (Mante et al., 2013) dataset is shown as a function of the variance explained by each PC. The highest variance PC is learned first during optimization at the lowest similarity score (bottom right of figure). A vertical slice through the figure shows the similarity score required to capture a specific PC. For example, to capture the PC at $10^{-2}$ requires a much lower similarity score when maximizing angular Procrustes versus CKA (light blue curve is below the red curve). *(c, d)* The reference dataset used as a target during optimization is the neural activity from (Siegel et al., 2015) FEF (panel *c*) and (Majaj et al., 2015) (panel *d*). *(e)* The neural data points are the same as in panels *c* and *d* (colored dots). The similarity scores at which PCs of this neural activity are learned, is well predicted by replacing neural activity with random Gaussian datasets that have a matching distribution of variances for each PC (black curves).

**Neural curves predicted by matched Gaussians.** Surprisingly, the optimization dynamics shown in these figures are well matched when the reference neural dataset is replaced by a new reference dataset that consists of random Gaussian numbers with variances for each principal component matched to the neural data (Figure 5e). The variance distribution of the reference dataset strongly determines when each principal component is captured during optimization.

### 4.3 THEORETICAL ANALYSIS OF CKA AND ANGULAR PROCRUSTES

We aim to understand the increased sensitivity of CKA scores to high variance principal components compared to angular Procrustes. Following Harvey et al. (2023), we use the Normalized Bures Similarity (NBS) metric as an intermediary metric between CKA and angular Procrustes. Specifically, we study the sensitivity to dimensions of different variances by analyzing the similarity scores between a neural dataset and a modified version of the same dataset when a single principal component (PC) is perturbed (Figure 6). We "perturb" a single PC of the neural dataset by first decomposing the dataset into its projections onto every PC and then replacing a single one of these projections by Gaussian random noise that is rescaled so the variance is unchanged.

**From NBS to CKA.** NBS and CKA mainly differ by a choice of matrix norm as shown in the following formulation (Harvey et al., 2023) (see appendix C.1 for details):

$$\text{CKA}(X, Y) = \frac{\|X^T Y\|_F^2}{\|X X^T\|_F \|Y Y^T\|_F} \qquad \text{NBS}(X, Y) = \frac{\|X^T Y\|_*}{\sqrt{\|X X^T\|_* \|Y Y^T\|_*}}$$

NBS quantifies similarity using the nuclear norm, which involves a sum of singular values, whereas CKA uses the Frobenious norm, which sums the square of the singular values. The additional square operation in CKA significantly increases the contribution of large variance components. We show this in theory by considering how CKA and NBS change when perturbing a single principal component of the data. The result is that CKA depends quadratically on the variance of the perturbed principal component, whereas NBS has a linear dependence (see proof in appendix C.2).

$$\text{CKA}(X, \tilde{X}_k) \approx \frac{\sum_{i \neq k} (\lambda_X^i)^2}{\sum_i (\lambda_X^i)^2} \qquad \text{NBS}(X, \tilde{X}_k) \approx \frac{\sum_{i \neq k} \lambda_X^i}{\sum_i \lambda_X^i}$$

where $\tilde{X}_k$ is equal to $X$ where only the $k^{\text{th}}$ principal component is modified, and $\lambda_X^i$ are the eigenvalues of $X X^T$.

We validate these results empirically by computing CKA and NBS scores between five different neural datasets and versions of these datasets when a single principal component is perturbed (see colored dots in Figure 6). As predicted by the theory, the NBS and CKA scores are well matched by, respectively, a linear and a quadratic function of the variance of the perturbed PC (see orange and blue curves).

**From angular Procrustes to NBS.** Harvey et al. (2023) showed that angular Procrustes is related to Normalized Bures Similarity (NBS) (Tang et al., 2020) by the arccosine function. We additionally normalize the angular Procrustes distance to obtain a score version, where 1 corresponds to perfect similarity.

$$\text{AngularProcrustesScore}(X, Y) = 1 - \arccos(\text{NBS}(X, Y)) * 2/\pi$$

The arccosine function decreases linearly around 0 with a slope of $-1$ and has a vertical asymptote in 1. When NBS is small, which corresponds to high variances of the perturbed PC in Figure 6, angular Procrustes is also linear. When NBS is around 1, which corresponds to low variances of the perturbed PC in Figure 6, angular Procrustes has increased sensitivity compared to NBS, as explained by the large slope of arccosine around 1.

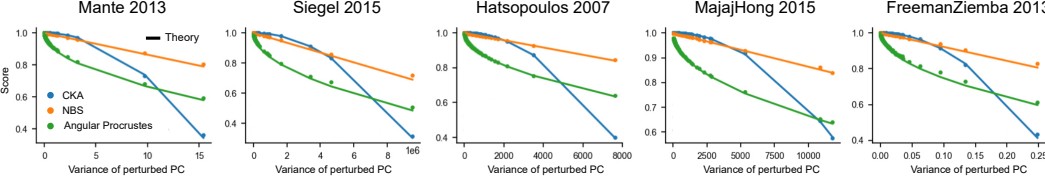

Figure 6: **The similarity score between five neural datasets and a modified version of these datasets when a single principal component is perturbed.** The colored dots show the empirical CKA, NBS, and angular Procrustes scores. The solid lines show the predictions of our theory. In particular, the CKA scores decrease quadratically with the variance of the perturbed principal component, whereas NBS scores decrease linearly. Angular Procrustes is related to NBS by the arccosine function, which explains its linear dependence to high variance PCs and its increased sensitivity to low variance PCs compared to NBS.

## 4.4 ARE METRICS MUTUALLY INDEPENDENT?

A natural question that arises is whether these different similarity metrics are independent of each other, or if they exhibit consistent relationships. To address this, we consider three possible relationships between any two given similarity metrics (Figure 7). **Independent:** The two metrics are independent. A high score on one metric offers no guarantee of a high score on the other. **Coupled:** The two metrics are tightly coupled. A high score on one metric implies a high score on the other, and vice versa. **Asymmetric:** One metric subsumes the other. A high score on the first metric guarantees a high score on the second, but not the other way around.

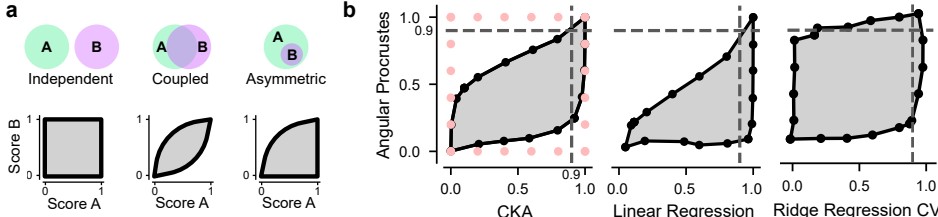

Figure 7: **We jointly optimized the values of both angular Procrustes and CKA or linear regression to illustrate the allowed ranges of both similarity scores** (gray region enclosed by the solid black lines). *(a)* Different categories of possible relationships between a similarity measure A and a similarity measure B. *(b)* Allowable ranges of similarity scores for a fixed reference dataset, illustrated here for the Siegel 2015 dataset. These ranges are estimated by jointly optimizing a pair of similarity measures to get as close as possible to different target values (shown with the pink dots forming a square on the first plot). If angular Procrustes has a high score of 0.9 (horizontal dashed line) then linear regression will have a value above this. In contrast, a high linear regression score of 0.9 (vertical dashed line) does not imply a high angular Procrustes score, and a wide range of angular Procrustes scores are possible (see appendix C.3 for details).

We jointly optimize multiple similarity scores to find their allowed ranges (Figure 7) and show that a **high angular Procrustes similarity implies a high CKA score, but not the converse.** A high value of angular Procrustes implies a high score for unregularized linear regression but linear regression that is regularized and cross-validated across experimental conditions can take independent values.

## 5    DISCUSSION

Our study reveals critical limitations of commonly used similarity measures for comparing models and neural datasets. While these measures offer a seemingly straightforward way to quantify representational alignment, our optimization-based approach shows that high similarity scores, particularly for CKA and linear regression, do not guarantee that synthetic datasets encode task-relevant information in a manner consistent with neural data. Specifically, we demonstrate that measures like CKA are heavily influenced by the top principal components of the data, often achieving high scores even when lower variance components, which might carry crucial task-related information, remain poorly captured.

The central aim of our work is to explore the question of what it means for a similarity score to be good, as well as what drives a high similarity score. Our findings demonstrate that the interpretation of these scores is highly dependent on the specifics of both the metric and the dataset. We argue that similarity scores require further interpretation before making any claim based on them. Here we show two concrete ways of doing so, that incorporate the context of the dataset and similarity measure, by optimizing synthetic datasets to determine (i) how strongly task variables are encoded in order to reach the score (Figure 3), and (ii) how many PCs need to be captured in order to reach the score (Figures 4 and 5).

**Limitations & future work.** Our study focused on differentiable, geometry-based similarity measures and their optimization dynamics. Future work should investigate how it extends to other types of measures. Additionally, when we study the optimization dynamics we are studying both the behavior of the optimization method and the behavior of the similarity measure. Future work should investigate the potential limitations of our gradient-based optimization method, and how to better untangle it from the behavior of the similarity measure.

Moving forward, our python package aims to standardize similarity measures, facilitating a more cumulative scientific approach by centralizing findings related to these measures, and enabling more integrated benchmarking and comparisons. There are many similarity measures we have not analyzed in this work, for example, even CKA has at least 12 variations, which are easily accessible in our package (Cloos et al., 2024). These similarity measures may, upon further investigation with the techniques we have proposed here, suffer from the same limitations we have demonstrated. Before using a similarity measure it is crucial to be aware of these limitations. Similarity package: https://github.com/nacloos/similarity-repository

## ACKNOWLEDGMENTS

We thank Laureline Logiaco for valuable discussions.

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

## A    ADDITIONAL RESULTS: DECODE ACCURACY OF TASK VARIABLES

Figures S1a and S1b show the decode accuracy of a linear classifier trained to decode task relevant variables for the Mante 2013 and the Siegel 2015 datasets (cross-validated across different conditions) as the similarity score increases. For both datasets, monkeys are shown a field of colored moving dots on each trial, and are required to attend to either color or motion information while ignoring the non-cued feature of the stimuli. Following Mante et al. (2013), we consider as the relevant task variables, the direction of motion of the dots, the color of the dots, the contextual cue, and the response of the monkey. We additionally binarize each of these variables for our decoding analysis. Before optimization, the synthetic datasets initially consisted of Gaussian noise and the decode accuracy was near the baseline chance level of 0.5 as expected for a binary classifier. We show results for the context task variable in Figure 3.

In Figure S2 we focus on a popular similarity measure, linear regression, and show how much task relevant information can be decoded from a synthetic dataset that is optimized to become increasingly more similar to the Siegel 2015 dataset. Even when linear regression is cross-validated and regularized, a high similarity score does not necessarily mean that task relevant information is encoded in a similar manner to neural data (leftmost panels).

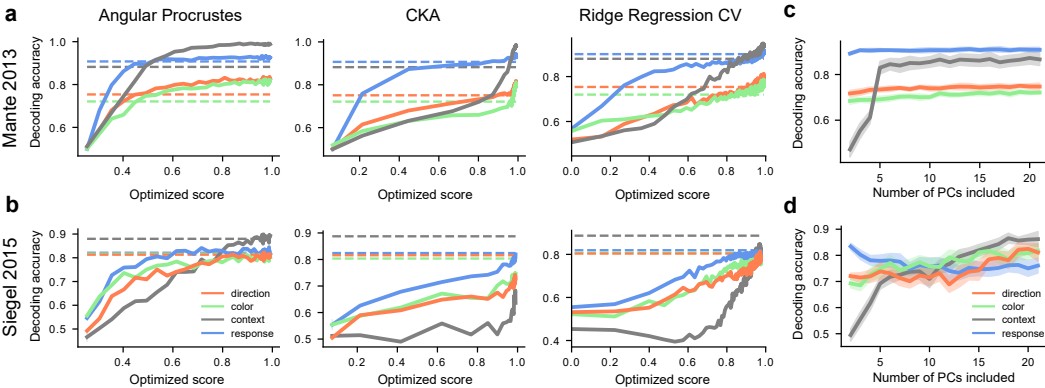

Figure S1: *(a, b)* **Decode accuracy for experimental variables versus similarity scores.** Decode is from synthetic data optimized towards greater similarity with the neural data from *(a)* Mante et al. (2013) and *(b)* Siegel et al. (2015). Horizontal dashed lines indicate the decode accuracy from the neural data. *(c, d)* Decode accuracy from neural data versus number of principal components included in the decode. Decode uses data from *(c)* prefrontal cortex from Mante et al. 2013 and *(d)* FEF from Siegel et al. 2015.

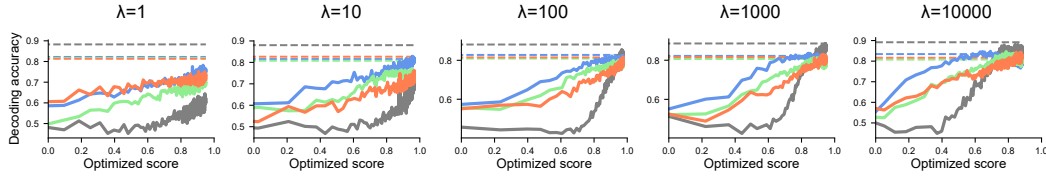

Figure S2: **Decode accuracy versus similarity scores when optimizing ridge regression scores with varying regularization strengths** $\lambda$. Regression scores are cross-validated across conditions. Horizontal lines show the decode accuracy from the neural data for the task relevant variables, and colored curves shows the corresponding decode accuracy on a synthetic dataset that is optimized to become increasingly more similar to the neural data. For some values of the ridge regularization parameter, even at the highest similarity scores near 1, the synthetic dataset does not encode task relevant information to the same extent as the neural data.

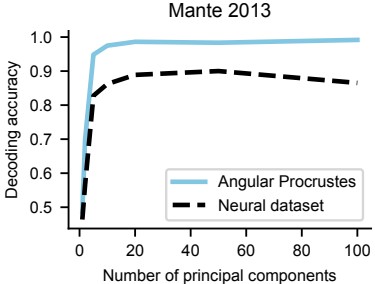

Figure S3: We test whether removing low-variance principal components from the neural dataset increases the decoding accuracy, in order to see if this "denoising" may be a potential mechanism for explaining the increased decoding accuracy in the optimized dataset compared to the reference neural dataset in Figure 3. We show results for decoding of the context task variables from the Mante 2013 dataset after projecting onto principal components 1 through N, where principal component 1 is taken to have the highest variance, and N increases along the x-axis of the figure. The solid blue line shows the decoding accuracy from the synthetic dataset optimized with Angular Procrustes when the similarity score reaches a value of 0.9. Although the decoding accuracy from the neural data increases slightly when the lowest variance components are removed, it does not match the decoding accuracy from the synthetic data. This suggests the improved decoding accuracy from the synthetic data is probably a more complex form of denoising and would require further investigation.

# B NEURAL DATASETS AND MODELS

## B.1 DATASETS

We analyzed neural data from five studies on nonhuman primates:

- Mante et al. (2013): Prefrontal cortex (PFC) electrode recordings during a contextual decision-making task involving colored moving dots. Activity is averaged across trials with the same condition (72 unique conditions) and averaged across 50 ms non-overlapping time bins during the 750 ms interval from 100 ms after dots onset to 100 ms after dots offset. The total number of neurons recorded is 727. To make optimization faster, we reduce the dimensionality of this dataset by keeping 99% of the variance, corresponding to the first 448 principal components. Data link[1]. The final dataset used in all the similarity analyses had size (time = 15, sample = 72, feature = 448).

- Siegel et al. (2015): Electrode recordings from multiple cortical regions during a contextual decision-making task involving colored moving dots, similar to Mante et al. (2013). We analyze data from the Frontal Eye Field (FEF) region during the stimulus period of the task. Activity is averaged across trials with the same condition (42 unique conditions) and averaged across 50 ms non-overlapping time bins from 0 to 500 ms after dots onset. A total of 1220 neurons were recorded. We reduce the dimensionality of this dataset by keeping 99% of the variance, corresponding to the first 278 principal components. The final dataset used in all the similarity analyses had size (time = 11, sample = 42, feature = 278).

- Hatsopoulos et al. (2007): Primary motor (M1) electrode recordings during a center-out reaching task. Simultaneous recordings were collected for 391 trials, 141 neurons. Activity is averaged across 50 ms non-overlapping time bins from 0 to 500 ms after movement onset. Data link[2]. The final dataset used in all the similarity analyses had size (time = 10, sample = 391, feature = 141).

- Majaj et al. (2015): Inferior temporal (IT) electrode recordings during object image presentations. Simultaneous recordings for 3200 trials, 168 neurons. Activity is averaged across all time steps. The final dataset used in all the similarity analyses had size (sample = 3200, feature = 168).

- Freeman et al. (2013): Primary (V1) and secondary (V2) visual area electrode recordings during texture and noise image presentations. Simultaneous recordings were collected for 135 trials,

---

[1] https://www.ini.uzh.ch/en/research/groups/mante/data.html
[2] https://datadryad.org/stash/dataset/doi:10.5061/dryad.xsj3tx9cm

205 neurons. Data is averaged across 150 ms non-overlapping time bins. The final dataset used in all the similarity analyses had size (time = 2, sample = 135, feature = 205).

We use the BrainScore[3] library (Schrimpf et al., 2020) for the Majaj et al. (2015); Freeman et al. (2013) datasets.

### B.2 RECURRENT NEURAL NETWORK (RNN) MODELS

**RNN architectures.** We show results for three commonly used RNN architectures: LSTMs (Hochreiter & Schmidhuber, 1997) and two choices for continuous time recurrent neural networks (CTRNNs), which differ by the position of the nonlinearity (Miller & Fumarola, 2012). A first alternative is given by the following equations.

$$\begin{cases} \tau\dfrac{dh}{dt} = -h + Wr + Bu + b \\ r = f(h) + \xi \end{cases}$$

where $u$ is the input, $h$ represents the membrane potential, $\xi$ is a Gaussian white noise, and $r$ is the firing rate produced by the model and is used to compare the model with the neural data. The second alternative is given by the following equation.

$$\tau\frac{dr}{dt} = -r + f(Wr + Bu + b) + \xi$$

We call it the LowPassCTRNN since it can be shown that its firing rate is a low-pass filter of the firing rate of the first architecture, which we call CTRNN (Miller & Fumarola, 2012). The RNNs are trained using supervised learning on simplified task inputs and outputs.

**Comparing task-optimized RNNs to neural datasets.** We train the RNNs on the tasks from Mante et al. (2013) and Siegel et al. (2015), and compare them to the respective neural datasets. We consider two neural datasets from prefrontal cortex (PFC) (Mante et al., 2013) and Frontal Eye Field (FEF) (Siegel et al., 2015) in monkeys performing an experimental task that required the animal to attend to either color or motion information while ignoring the non-cued feature of the stimuli. On each trial, a field of colored moving dots is shown. Monkeys are given a cue at the beginning of the trial to determine whether the dots in the stimulus are moving left vs right, or are red vs green. The monkey reported its choice with a saccade to one of two visual targets. In both datasets, we analyzed neural activity taken when the dot stimulus was presented. In panel *a* of Figure 2 neural activity is visualized in a low-dimensional space capturing task-relevant dynamics using the targeted dimensionality reduction method from Mante et al. 2013. Each curve shows the average neural activity for a different experimental condition. See Mante et al. 2013 for a detailed description of the analysis. This visualization highlights features of the data but the similarity scores were computed using the firing rates from the electrode recordings before any dimensionality reduction. In panel *b* of Figure 2 neural firing rates for two example neurons are shown with the colors denoting average.

In Figure 2 *c, d*, we compare model-data similarity scores to two baseline scores. The Neuron-split baseline score is obtained by dividing the neurons from a single dataset into disjoint sets and then comparing. If the model-data scores are equal to the neuron-split scores this indicates that model activity is indistinguishable from the neural activity of other recorded neurons. The Condition-average baseline score is obtained by averaging the neural activity along the trial and condition dimensions and comparing it to the original dataset. Each neuron in this condition-averaged dataset still has a unique time-varying firing rate. This strong baseline shows the similarity to the original data that one can obtain by only keeping the condition-independent neural dynamics.

## C SIMILARITY MEASURES

### C.1 DEFINITIONS

Several methods have been proposed to measure similarity between models and neural data (see (Sucholutsky et al., 2024), (Klabunde et al., 2023) for comprehensive reviews). While some efforts have been made to characterize these metrics mathematically, for instance, by examining their

---

[3]https://github.com/brain-score/vision

invariance properties, clear guidance in interpreting similarity scores for a given scenario remains limited. To address this gap, we introduce a novel method for analyzing the specific aspects of data prioritized by different similarity metrics. Our method leverages the differentiability of these metrics and is applicable to a wide range of measures. We focus here on three commonly used metrics and their variants, linear regression (Yamins et al., 2014; Schrimpf et al., 2018), Centered Kernel Alignment (CKA) (Kornblith et al., 2019), and Procrustes distance (Williams et al., 2021; Ding et al., 2021). These methods quantify similarity based on the goodness of fit after aligning the representations. This alignment transformation allows for greater flexibility by making similarity measures invariant to specific transformations. For example, instead of demanding a one-to-one mapping of neurons between datasets, these methods can accommodate scenarios where neurons in one dataset correspond to linear combinations of neurons in the other.

Consider two datasets $X$ and $Y$ with shape (sample, feature) and mean-centered columns. Datasets with dynamics are shaped from (time, sample, feature) to (time*sample, feature).

Assuming $X$ as our reference dataset (e.g., neural data), linear regression seeks the optimal linear mapping $B$ to predict $X$ from $Y$. We use the $R^2$ coefficient to evaluate goodness of fit and ridge regularization with parameter $\lambda = 100$, as well as 5-fold cross-validation. Note that we apply cross-validation after reshaping the data from (time, sample, feature) to (time*sample, feature).

$$B^* = \arg\min_{B} \|X - YB\|_F^2 + \lambda\|B\|_F$$

$$R_{LR}^2 = 1 - \frac{\|X - YB^*\|_F^2}{\|X\|_F^2}$$

It's important to note that this score is not symmetric. Applying linear regression on $Y, X$ instead of $X, Y$ may yield significantly different scores. This asymmetry arises because one dataset might be highly predictive of the other, while the reverse might not hold true.

Unlike linear regression, CKA and Procrustes are symmetric, meaning the metric yields the same result whether applied to X, Y or Y, X. Moreover, they exhibit a different class of invariance. While linear regression is invariant under invertible linear transformations, CKA and Procrustes are invariant under orthogonal linear transformations. This stricter invariance to orthogonal transformations potentially reduces sensitivity to noise (Kornblith et al., 2019).

CKA measures the correlation between the kernels of two datasets $X$ and $Y$ (Kornblith et al., 2019). We consider here linear CKA where the kernels are $XX^T$ and $YY^T$.

$$\text{CKA}(X, Y) = \frac{\langle \text{vec}(XX^T), \text{vec}(YY^T) \rangle}{\|XX^T\|_F \|YY^Y\|_F} = \frac{\|X^TY\|_F^2}{\|XX^T\|_F \|YY^T\|_F}$$

CKA scores range from 0 to 1, with 1 indicating perfect similarity. We also consider the arccosine of CKA, a metric satisfying the axioms of a distance metric (e.g., the triangle inequality) with 0 signifying perfect similarity (Williams et al., 2021). This metric, also known as Angular CKA (Lange et al., 2022), is then normalized to a similarity score between 0 and 1 for direct comparison with other scoring methods. This normalization involves dividing the angular distance by $\pi/2$ and subtracting the result from 1, ensuring the measure increases with similarity.

$$\text{AngularCKAScore}(X, Y) := 1 - \frac{\arccos(\text{CKA}(X, Y))}{\pi/2}$$

Procrustes distance provides another approach for quantifying similarity (Williams et al., 2021; Ding et al., 2021). This metric identifies the optimal orthogonal alignment to maximize the correlation between $X$ and $Y$:

$$\max_{Q \in \mathcal{O}} \frac{\langle X, QY \rangle}{\|X\|_F \|Y\|_F}$$

where $\mathcal{O}$ is the group of orthogonal linear transformations. An angular distance metric can be obtained by taking the arccosine of this quantity (Williams et al., 2021).

As shown in Harvey et al. (2023), Procrustes distance is closely related to Normalized Bures Similarity (NBS) (Tang et al., 2020), with Procrustes distance equating the arccosine of NBS. NBS is defined as:

$$\text{NBS}(X, Y) = \frac{\|X^T Y\|_*}{\sqrt{\|XX^T\|_* \|YY^T\|_*}}$$

This definition resembles CKA but utilizes the nuclear matrix norm instead of the Frobenius matrix norm. The Frobenius norm, denoted by $\|A\|_F$ for a matrix $A$, is calculated as the square root of the sum of squared singular values. The nuclear norm, $\|A\|_*$, is simply the sum of singular values. The implications of this difference in matrix norms for the weighting of principal components by CKA and NBS are further explored in C.2.

Similar to the score version of CKA distance, we define a score version of Procrustes distance, ranging from 0 to 1, where 1 represents perfect similarity.

$$\text{AngularProcrustesScore}(X, Y) := 1 - \frac{\arccos(\text{NBS}(X, Y))}{\pi/2}$$

## C.2 THEORETICAL ANALYSIS OF CKA AND NBS

Our results show that similarity in the high variance components seem to dominate the Centered Kernel Alignment (CKA) score. In contrast, while the high variance components still have an overall larger impact than lower variance components, metrics like Normalized Bures Distance (NBS) seem to be more sensitive to changes in lower variance components relative to CKA. We explain this difference by showing, in theory, how CKA and NBS changes when perturbing a single principal component of the data. The result is that CKA depends quadratically on the variance of the perturbed principal component, whereas NBS has a linear dependence. We start the derivation from the matrix norm definitions of CKA and NBS (Harvey et al., 2023).

$$\text{NBS}(X, Y) = \frac{\|X^T Y\|_*}{\sqrt{\|XX^T\|_* \|YY^T\|_*}}$$

$$\text{CKA}(X, Y) = \frac{\|X^T Y\|_F^2}{\|XX^T\|_F \|YY^T\|_F}$$

Let $X$ be a dataset, and $Y$ be a perturbed version of $X$, denoted $\tilde{X}_k$, where only the $k^{\text{th}}$ principal component is modified. Specifically, define $Y$ such that its $j^{\text{th}}$ left singular vector $u_Y^j$ is equal to the $j^{\text{th}}$ left singular vector $u_X^j$ of $X$ for all $j \neq k$, and define $u_Y^k$ to be a perturbation of $u_X^k$ that preserves variance. Here we randomly sample $u_Y^k$ from a Gaussian distribution with the same variance as the variance of $u_X^k$ so that $\sigma_Y^k = \sigma_X^k$. The right singular vectors of $Y$ are the same as the right singular vectors of $X$. Intuitively this perturbation affects only the projection of the data along the $k^{th}$ principal component.

We first show how $\text{CKA}(X, \tilde{X}_k)$ depends on the variance of the perturbed principal component. As shown in (Kornblith et al., 2019), CKA can be rewritten in terms of dot products of the left singular vectors.

$$\text{CKA}(X, Y) = \frac{\sum_{i,j} \lambda_X^i \lambda_Y^j \langle u_X^i, u_Y^j \rangle^2}{\sqrt{\sum_i (\lambda_X^i)^2} \sqrt{\sum_j (\lambda_Y^j)^2}}$$

where $X = U_X \Sigma_X V_X^T$, $Y = U_Y \Sigma_Y V_Y^T$ are the SVD decompositions of $X$ and $Y$ respectively, and $\lambda_X^i = (\sigma_X^i)^2$, $\lambda_Y^i = (\sigma_Y^i)^2$. Since $u_Y^j = u_X^j$ for all $j \neq k$ and since $U_X$ is an orthogonal matrix, the dot products $\langle u_X^i, u_Y^j \rangle$ are equal to 0 for all $j \neq i$ whenever $j \neq k$. The dot products $\langle u_X^i, u_Y^k \rangle$ are not guaranteed to be zero since randomly sampling $u_Y^k$ doesn't guarantee to preserve the orthogonality with the other left singular vectors. However, when the number of samples in the data is large the projection $u_Y^k$ on the other singular vectors will be relatively small. With the assumption that $\langle u_X^i, u_Y^k \rangle \approx 0$, CKA can be written as:

$$\text{CKA}(X, \tilde{X}_k) \approx \frac{\sum_{i \neq k} (\lambda_X^i)^2}{\sum_i (\lambda_X^i)^2}$$

This shows that CKA scores between the original data and the perturbed data depend quadratically on the variance of the perturbed principal component. As shown in Figure 6, our theoretical approximation closely matches simulations.

As opposed to CKA, NBS cannot be directly rewritten as a sum of left singular vector dot products. However, we show that NBS reduces to a simple form when $Y$ is defined as $X$ perturbed along a single principal component and when $\langle u_X^i, u_Y^k \rangle \approx 0$ is assumed. We start by expressing the nuclear norm in the numerator of NBS in terms of the SVD decomposition of $X$ and $Y$.

$$\|X^T Y\|_* = \|V_X \Sigma_X U_X^T U_Y \Sigma_Y V_Y^T\|_* = \|\Sigma_X U_X^T U_Y \Sigma_Y\|_*$$

The individual entries of the product of matrices inside the nuclear norm corresponds to the dot products between the left singular vectors of $X$ and $Y$ weighted by the corresponding singular values.

$$[\Sigma_X U_X^T U_Y \Sigma_Y]_{ij} = \sigma_X^i \sigma_Y^j \langle u_X^i, u_Y^j \rangle$$

By definition of $Y \equiv \tilde{X}_k$,

$$\sigma_X^i \sigma_Y^j \langle u_X^i, u_Y^j \rangle = \begin{cases} (\sigma_X^i)^2 \delta_{ij} & \text{if } j \neq k \\ \sigma_X^i \sigma_X^k \langle u_X^i, u_Y^k \rangle & \text{if } j = k \end{cases}$$

With our assumption that $\langle u_X^i, u_Y^k \rangle \approx 0$ and the nuclear norm property $\|A\|_* = \text{Tr}\left[\sqrt{A^T A}\right]$, we find:

$$\|X^T Y\|_* \approx \sum_{i \neq k} (\sigma_X^i)^2$$

Since our perturbation preserves the variance, we have $\|XX^T\|_* = \|YY^T\|_* = \sum_i (\sigma_X^i)^2$. We finally obtain the following approximation, which explicitly reveals the linear dependence of NBS on the variance of the principal components.

$$\text{NBS}(X, \tilde{X}_k) \approx \frac{\sum_{i \neq k} (\sigma_X^i)^2}{\sum_i (\sigma_X^i)^2} = \frac{\sum_{i \neq k} \lambda_X^i}{\sum_i \lambda_X^i}$$

## C.3 JOINT OPTIMIZATION OF SIMILARITY MEASURES

Figure 7 shows how two metrics relate by jointly optimizing for both metrics. Given a score for the first metric and a score for the second metric, we ask whether it is possible to find a single synthetic dataset that would produce these two similarity scores. Specifically, for a given reference dataset $X$, similarity measures $A$ and $B$, and respective targets $\text{target}_A$ and $\text{target}_B$, we optimize a synthetic dataset $Y$ to minimize the sum of absolute errors:

$$\min_Y \left[ |\text{score}_A(X, Y) - \text{target}_A| + |\text{score}_B(X, Y) - \text{target}_B| \right]$$

We chose the absolute error instead of the squared error as we find in practice that it gives scores that are closer to their targets.

This analysis gives an approximation of the range of scores that two metrics can jointly take. For example, if two metrics are completely independent, then any point on the $[0, 1] \times [0, 1]$ square is reachable (see third plot of Figure 7b for an example). However, if there exists a one-to-one mapping between the two metrics, then the scores are constrained to lie on a one-dimensional curve.

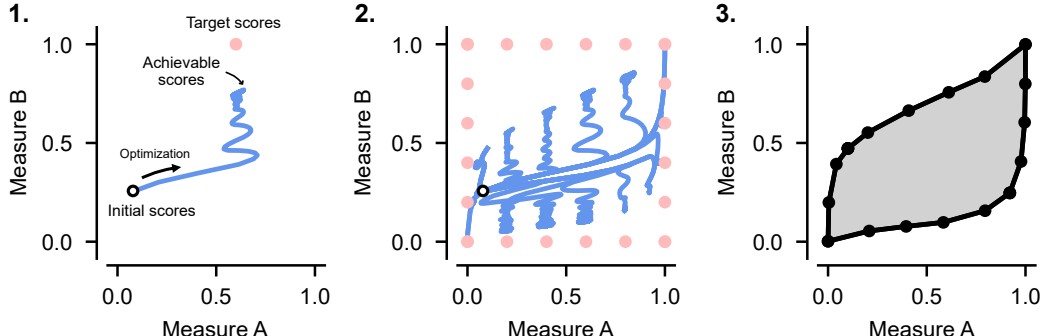

Figure S4: **Computing the allowed ranges between two similarity scores by optimizing synthetic data towards a grid of target similarity scores.** *(1)* Given a synthetic dataset $Y$ and a reference dataset $X$ we attempt to simultaneously achieve the desired target scores for similarity measures A and B (pink circle). To do this we optimize $Y$ with the Adam optimizer by differentiating through the scores for similarity measure A, $\text{score}_A(X, Y)$, and B, $\text{score}_B(X, Y)$, to minimize a sum of absolute error loss function between the current and target similarity scores. As $Y$ is optimized the scores for measures A and B trace out the curve shown in blue. Note that the final achievable scores for the two similarity measures approach but, in this case, do not reach the target scores. *(2)* This optimization procedure is repeated for many different target scores (pink dots) and the intermediate scores are recorded (blue curves). *(3)* The edges of the blue curves are marked with black dots and approximate a boundary of achievable scores as shown in gray.

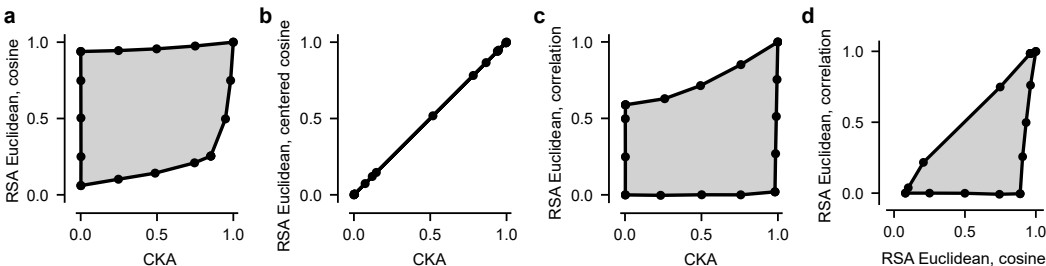

Figure S5: **We analyze different variations of Representational Similarity Analysis (RSA) (Kriegeskorte et al., 2008a) and CKA by jointly optimizing the similarity measures to approximate the allowable regions of scores.** *(a)* We compare CKA and a commonly used version of RSA that uses the Euclidean distance to compute the Representational Dissimilarity Matrices (RDMs) and cosine similarity to compare the RDMs. We find that these two similarity measures are mostly independent, where a high score for one measure doesn't necessarily imply a high score for the other. *(b)* We confirm the equivalence between CKA and RSA when centering the Euclidean RDMs before computing the cosine similarity, as shown by Williams (2024). *(c)* We compare CKA to another common variation of RSA that computes the correlation between the RDMs instead of the cosine similarity. We find that these two measures are also mostly independent. *(d)* We compare two RSA variants together and show that changing the RDM comparison method makes the RSA measures non-equivalent.

