# OpenReview forum: "Differentiable Optimization of Similarity Scores Between Models and Brains"
_ICLR.cc/2025/Conference — ICLR 2025 Poster_

### Official Review · Reviewer_w1zm · 2024-10-31

**Soundness:** 2
**Presentation:** 2
**Contribution:** 2
**Rating:** 6
**Confidence:** 4

**Summary:**

This paper investigates the properties of several similarity metrics in various neural activity datasets. The goal of similarity metrics is to quantify how well models of brain align with neural data. However, there are inconsistencies across different metrics, i.e., some metrics score high while others score low. This paper aims to address this inconsistency problem and propose a model-agnostic synthetic dataset optimization to analyze the properties of similarity metrics. The optimization dynamics in numerical experiments reveal that there is no single metric that is universally applicable for all dataset since the concept of a good score is highly dependent on the dataset. Additionally, the authors provide a python package that includes various similarity metrics.

**Strengths:**

* The problem is well-formulated in the introduction and clearly illustrated in Figure 2.
* Numerical experiments are presented clearly for the reader.
* The published code is well-structured, enhancing reproducibility.
* The observations made in Figure 3 are interesting (that some scores are good for some datasets while they are bad for other datasets).

**Weaknesses:**

The paper has some clarity and novelty issues in my opinion. Please see the points below and the questions section.

* One premise stated in the abstract is that the paper offers a theoretical analysis to show how similarity metrics are dependent on the principal components of the dataset. However, this premise appears weak to me because: (i)  it does not seem to be a novel analysis but rather a predictable outcome of using Frobenius and nuclear norms in metrics CKA and NBS; (ii) the assumption $\langle u_X^i, u_Y^i\rangle \approx 0$  is introduced without sufficient context and is unclear; and (iii) the assumption is said to hold with large sample sizes, validated in a numerical experiment, yet I did not see a clear mention of dataset sizes in the paper. Including more details on the datasets and clarifying the underlying assumptions would be helpful.

* The introduction and related work section suggest that prior research lacks practical guidance on metric selection given a dataset. However, I am uncertain if this paper proposes such a guidance. Suppose I have a neural dataset $X$ and model representations $Y$ to compare. How should I choose the most suitable metric based on this paper? My understanding is that I can optimize a synthetic dataset $Z$ using various metrics, observe the optimization dynamics, and then choose a similarity metric for $X$ and $Y$. Is that correct? I am asking this since I am struggling to understand how this paper offers a method for selecting an appropriate metric for a given task, if indeed it is promising.

* The claim between lines 267-270 is not detailed enough in the paper. The authors mention testing a hypothesis, but they simply state "we tested this hypothesis ... but this did not change the results" without further context. The results for that is not shared in the paper. Including these results in the appendix would enhance the paper's clarity.

* The joint optimization method in Section 4.4 and Appendix C.3 is unclear, as the details on experiments in this section are sparse. I think the paper can benefit from more details.

* The term "Proof" in Appendix C.2 and Section 4.3 seems a bit strong without an accompanying theorem or lemma, especially since the assumption is only noted in the appendix. Revising the word "proof" might be appropriate.

**Minor Comments**

* The sentence in line 417 is repeated; it was already mentioned that Williams et al. (2021) advocate taking the arccos of CKA to align with distance metric axioms.

* Appendix B.1 could provide more dataset details rather than referring readers to other works. Including information such as dataset dimensions and data collection methods would be helpful.

**Questions:**

* What is the reason for using ridge regularization in the $R^2$ definition (in the numerator in line 208)? In that case, the numerator will not be the residual square and I do not see the rationale behind this.

* Why is line 512 specifically bold-faced, but not the next one? According to Figure 7, the relation that high value of angular Procrustes implies a high score for linear regression appears more established compared to the relation of angular Procrustes and CKA scores.

* Given the findings, the main takeaway seems to be that similarity metrics are highly sensitive to different data aspects and may be mutually independent. How, then, would the authors suggest selecting the best similarity metric for a given dataset?

**Post rebuttal comment:** I appreciate the authors' detailed responses and the revisions made to the paper. These changes have significantly improved its clarity in my opinion. Considering the positive feedback from other reviewers as well, I am pleased to raise my score to 6.

---

> ### Author Response · Authors · 2024-12-02
>
> Thank you for your assessment and suggestions. They were quite useful in revising the paper to better communicate the goals of our study. We address your concerns/suggestions below.
>
> > One premise stated in the abstract is that the paper offers a theoretical analysis to show how similarity metrics are dependent on the principal components of the dataset. However, this premise appears weak to me because: (i) it does not seem to be a novel analysis but rather a predictable outcome of using Frobenius and nuclear norms in metrics CKA and NBS; (ii) the assumption is introduced without sufficient context and is unclear; and (iii) the assumption is said to hold with large sample sizes, validated in a numerical experiment, yet I did not see a clear mention of dataset sizes in the paper. Including more details on the datasets and clarifying the underlying assumptions would be helpful.
>
> It is true that the quadratic and linear dependence of CKA and NBS sounds intuitive from the difference between the Frobenius and nuclear norm. However, it is not straightforward to relate it to our empirical results showing the dependence of the scores to the principal components of one of the two datasets being compared. In particular, the numerator in the matrix norm formulation of CKA and NBS is the norm of the product $X^T Y$, meaning it is the sum of the singular values of $X^T Y$ (squared for CKA). However, we want to analyze the dependence with respect to the singular values of only one of the two matrices. In our analysis, we show how to do that using the formulation of CKA in terms of a weighted sum of inner products between left singular vectors from Kornblith et al. (2019). NBS doesn’t have such an alternative formulation but we show that with our assumptions on the perturbation on $X$, we can derive a similar expression. So even though the results are expected, our contribution is to explicitly show the dependence on the principal components of one of the two datasets and to quantitatively validate it on neural datasets.
>
> Thanks for your suggestions to clarify this section. We added additional details to the revised paper.
>
>
>
> > The introduction and related work section suggest that prior research lacks practical guidance on metric selection given a dataset. However, I am uncertain if this paper proposes such a guidance. Suppose I have a neural dataset and model representations to compare. How should I choose the most suitable metric based on this paper? My understanding is that I can optimize a synthetic dataset using various metrics, observe the optimization dynamics, and then choose a similarity metric for and . Is that correct? I am asking this since I am struggling to understand how this paper offers a method for selecting an appropriate metric for a given task, if indeed it is promising.
>
> > Given the findings, the main takeaway seems to be that similarity metrics are highly sensitive to different data aspects and may be mutually independent. How, then, would the authors suggest selecting the best similarity metric for a given dataset?
>
> We have updated the abstract and main text to hopefully remove the confusion about our motivations. To clarify, the specific questions that motivated this paper are:
> * What constitutes a “good” score? See **Figure 3**.
> * What drives a high similarity score? See **Figures 4, 5** and the analytic results in **section 4.3** as well as **Figure 6**.
> * Does a high similarity score for one measure imply a high score for the others? See **Figure 7**.
>
> At this point in time we do not wholeheartedly recommend any of the current similarity measures without some reservations. That being said, not all similarity measures have the same drawbacks and we see angular Procrustes as a reasonable default. But see caveats in, for example, Ostrow et al. NeurIPS 2023 and Khosla & Williams UniReps 2023.
>
> In terms of practical guidance, we advise to not just report similarity scores as our findings show that scores by themselves are hard to interpret and their meaning can greatly vary depending on the details of the dataset and the similarity measure. We argue that similarity scores require further interpretation before making any claim based on them. Here we show two concrete ways of doing so, that incorporate the context of the dataset and similarity measure, by optimizing synthetic datasets to determine:
> 1. How strongly task variables are encoded in order to reach the score (**Figure 3**).
> 2. How many PCs need to be captured in order to reach the score (**Figures 4 and 5**).

---

> ### Author Response · Authors · 2024-12-02
>
> > The claim between lines 267-270 is not detailed enough in the paper. The authors mention testing a hypothesis, but they simply state "we tested this hypothesis ... but this did not change the results" without further context. The results for that is not shared in the paper. Including these results in the appendix would enhance the paper's clarity.
>
> Thanks for your suggestion. We added a new figure to support the claim (**Supplementary Figure S3**).
>
> > The joint optimization method in Section 4.4 and Appendix C.3 is unclear, as the details on experiments in this section are sparse. I think the paper can benefit from more details.
>
> We added more details to **Appendix C.3** and a new figure to explain the procedure (**Supplementary Figure S4**).
>
> > The term "Proof" in Appendix C.2 and Section 4.3 seems a bit strong without an accompanying theorem or lemma, especially since the assumption is only noted in the appendix. Revising the word "proof" might be appropriate.
>
> We agree the term “proof” may be considered too strong here. We revised the title for this section.
>
> > What is the reason for using ridge regularization in the definition (in the numerator in line 208)? In that case, the numerator will not be the residual square and I do not see the rationale behind this.
>
> Thanks for pointing this out. This was a typo in the manuscript, which is now corrected. The ridge regularization is only used to find the best linear mapping and not to evaluate it with $R^2$.
>
> > Why is line 512 specifically bold-faced, but not the next one? According to Figure 7, the relation that high value of angular Procrustes implies a high score for linear regression appears more established compared to the relation of angular Procrustes and CKA scores.
>
> There are commonly used versions of both angular Procrustes and CKA and so the bolded statement is of general interest. We did not bold the statement concerning linear regression as we felt there are too many variations used in practice. Our analyses of linear regression illustrate the potential problems with even cross-validated and regularized versions of this similarity measure but we encourage practitioners to apply our method to their unique setup and choices for cross-validation, regularization, and dataset.
>
> In light of these clarifications, would you consider increasing your score for our paper? If not, could you let us know any additional changes you would like to see in order for this work to be accepted?

---

### Official Review · Reviewer_c7uz · 2024-11-02

**Soundness:** 3
**Presentation:** 3
**Contribution:** 3
**Rating:** 6
**Confidence:** 3

**Summary:**

This paper aims to compare how different similarity measures such as CKA and linear regression behave, based on both prior theoretical work as well as by optimizing synthetic data through Adam to become more similar (under some similarity measure, e.g. CKA) to a reference neural dataset. The paper analyzes what properties a synthetic dataset can be expected to have (e.g. with respect to decodability of task relevant variables) at various levels of similarity to a reference dataset under a range of similarity measures.

**Strengths:**

- Although optimizing a set of features to become more similar to neural data has been done (e.g. optimizing neural network models of the brain), specifically optimizing a synthetic dataset to in order to gain insight into how similarity measures behave, especially at various intermediate levels of similarity, is novel.
- Most discussions of similarity measures have focused on the special case where the similarity score is 1 (for example, what happens when response profiles X and Y are equivalent under some similarity measure such as CKA), so discussion of how intermediate values behave for different measures is a good contribution, especially since we are often dealing with intermediate levels of similarity in practice, e.g. when comparing models to brain data.
- CKA and linear regression are widely used methods of measuring similarity, so this paper can potentially be useful to many researchers comparing models to brain data.

**Weaknesses:**

- In this paper, the regularization level for Ridge Regression is fixed to some chosen level (and the authors do consider results for different fixed levels of lambda). However it seems to me that, because of the probability of overfitting in high dimensional data settings, it is generally preferable to tune the ridge penalty through some cross-validation method (searching over a range of possible alpha values) such as k-fold so as to select the lambda that will maximize generalization performance on the chosen data.
- Linear regression is only done in one direction, from the model to the reference neural dataset. This is good to know about, but it also would be useful to see what happens when linear regression is done in both directions, i.e. if synthetic data is optimized so that it predicts the brain and the brain predicts the data as well.
- RSA is also widely used as a similarity measure, but is not mentioned at all in the paper. It would be very useful if the paper included an analysis of RSA, especially since RSA is mathematically very closely related to linear CKA, but the formulas for RSA and linear CKA are not identical. It would be good to therefore have an analysis of the relationship between these two methods, as well as empirical simulations showing how the intermediate values compare for RSA and CKA (just as the authors did for other methods, like comparing CKA to NBS).
- While many parts of the paper are clearly written, Figures 5 and 6 were hard for me to understand, and there was not much explanation in either the caption or main text. See questions below.
- While I understand the intended application of these results is to help researchers better understand similarity scores when comparing models to brains, the paper title seems a bit misleading, since it gives the impression that the paper is optimizing the similarity between an actual ANN model and the brain, whereas what is actually done here is optimizing similarity scores between a randomly initialized matrix and the brain features. Perhaps the title doesn't need to be fixed, but initially the title gave me a different idea of what the paper was going to do.

**Questions:**

In Figure 5, what does PC explained variance mean? What is the PC threshold? (I'm also not sure which PC we are talking about here - is it the first, largest PC?) Why is it that the score to reach PC threshold is *larger* for a smaller PC explained variance? Shouldn't explaining less variance require a smaller score?

In Figure 6, what does it mean to perturb a single PC? How much do you perturb that PC? (it isn't stated, but I assume that how much you perturb it is very important for what the resulting similarity score should be).

---

> ### Author Response · Authors · 2024-12-02
>
> Thank you for your positive assessment and suggestions. We address your concerns/suggestions below.
>
> > Most discussions of similarity measures have focused on the special case where the similarity score is 1 (for example, what happens when response profiles X and Y are equivalent under some similarity measure such as CKA), so discussion of how intermediate values behave for different measures is a good contribution, especially since we are often dealing with intermediate levels of similarity in practice, e.g. when comparing models to brain data.
>
> We completely agree. This is a great comment and a point we now emphasize in the paper.
>
> > In this paper, the regularization level for Ridge Regression is fixed to some chosen level (and the authors do consider results for different fixed levels of lambda). However it seems to me that, because of the probability of overfitting in high dimensional data settings, it is generally preferable to tune the ridge penalty through some cross-validation method (searching over a range of possible alpha values) such as k-fold so as to select the lambda that will maximize generalization performance on the chosen data.
>
> Tuning the ridge penalty with cross-validation (CV) is indeed commonly used to select a ridge regression metric. The method we propose is complementary and addresses a different question. Once we have a given similarity measure (e.g. ridge regression with a specific value of lambda obtained with CV), how can we interpret and validate the scores it produces? In particular, is it possible to reach a high similarity score without capturing the relevant task variables or capturing only the first PCs?
>
> > Linear regression is only done in one direction, from the model to the reference neural dataset. This is good to know about, but it also would be useful to see what happens when linear regression is done in both directions
>
> We focused on illustrating our method on commonly used similarity measures, and linear regression as a metric is mostly used in one direction. Our goal is to demonstrate the usefulness of our approach and give insights on commonly used similarity measures. We additionally provide tools to the community with, for example, open source code, to analyze and better understand present and future similarity measures.
>
> > RSA is also widely used as a similarity measure, but is not mentioned at all in the paper. It would be very useful if the paper included an analysis of RSA, especially since RSA is mathematically very closely related to linear CKA, but the formulas for RSA and linear CKA are not identical. It would be good to therefore have an analysis of the relationship between these two methods, as well as empirical simulations showing how the intermediate values compare for RSA and CKA (just as the authors did for other methods, like comparing CKA to NBS).
>
> RSA is indeed widely used and interesting to consider as there are many existing variants. We have now included a new figure comparing common variants of RSA with CKA (**Supplementary Figure S5**). In particular, our new results confirm the equivalence between RSA with Euclidean RDMs and centered cosine similarity (see Williams UniReps 2024). However, they also show that other variations of RSA are almost independent of CKA, and that they might not even be equivalent to each other (as illustrated by comparing RSA with Euclidean RDMs using either cosine similarity or correlation to compare the RDMs).

---

> > ### Author Response · Authors · 2024-12-02
> >
> > > In Figure 5, what does PC explained variance mean? What is the PC threshold? (I'm also not sure which PC we are talking about here - is it the first, largest PC?) Why is it that the score to reach PC threshold is larger for a smaller PC explained variance? Shouldn't explaining less variance require a smaller score?
> >
> > Thank you for these helpful comments. We have updated the paper to clarify.
> >
> > * The “PC explained variance” is the variance explained by each principal component of the neural dataset used as the target dataset for the optimization (the particular neural dataset is written in each figure). We have now changed the x-axis labels in **Figure 5** to “Variance explained by each PC”.
> >
> > * The PC threshold is just a convenient way of summarizing the observation from **Figure 4** that as the optimized score increases, each principal component (PC) of the target dataset is better captured (see **section 3.2** for the mathematical description of “captured”), and at some score (the score to reach PC threshold) about half of that PC’s variance is captured by the optimized dataset. Stated more precisely, the PC threshold for principal component j (shown in **Figure 5a** for PC 1) is the centerpoint between the maximum and initial R2 value for principal component j as the synthetic dataset is optimized, where j = 1, 2, 3, etc. Each principal component of the target dataset will likely be captured at a different similarity score and so the score to reach PC threshold will be different for each PC.
> >
> > * As synthetic datasets are optimized to become more similar to a target dataset, the highest variance PCs of the target dataset are captured first (see **Figure 4b**), and so the score to reach PC threshold is smaller for these larger variance components (note that the score is initially near the minimum of 0 and increases towards the maximum of 1 as optimization progresses). In other words, a higher score is required to capture smaller variance components and so the PC threshold for these lower variance principal components is not reached until a higher score is achieved.
> >
> > > In Figure 6, what does it mean to perturb a single PC? How much do you perturb that PC? (it isn't stated, but I assume that how much you perturb it is very important for what the resulting similarity score should be).”
> >
> > We “perturb” a single PC of the neural dataset by first decomposing the dataset into its projections onto every PC and then replacing a single one of these projections by Gaussian random noise that is rescaled so the variance is unchanged. We described this procedure in **appendix C.2** but have now clarified this in the main text as well.

---

### Official Review · Reviewer_CM3i · 2024-11-03

**Soundness:** 4
**Presentation:** 3
**Contribution:** 3
**Rating:** 8
**Confidence:** 4

**Summary:**

The paper aims to compare various similarity measures by optimizing randomly initialized datasets to various neural recording datasets. The authors find that some measures such as CKA can have high scores without sufficiently encoding task relevant information. The paper then investigates how much of a dataset needs to be captured before a certain score is achieved. It finds that some of the measures that have high scores without encoding task-relevant information also are most sensitive to high variance principal components. The authors complete theoretical and perturbation experiments to validate this hypothesis.

**Strengths:**

1. Very Clear writing, easy to see what analysis is being done and why.
2. Evaluating in a model agnostic way puts the focus on the measures and leads to a better understanding of the relevant differences for completing model-brain comparisons.
3. This analysis is fundamental to the field. Understanding what aspects lead to a high similarity score is extremely important to guide development of new models and to properly apply the modeling results to the brain.

**Weaknesses:**

1. Ridge-Regression seems to be the most widely-used measure in the field although most of the comparisons focus on CKA vs Angular Procrustes. Would be nice to see more commentary on this especially in Fig. 7 where it seems independent from angular Procrustes.
2. From the start of the paper it seems like it will answer the question: "What metrics should guide the development of more realistic models of the brain?" The discussion seems to attempt to avoid this question: "Our findings demonstrate that the interpretation of these scores is highly dependent on the specifics of both the metric and the dataset. We do not claim that one metric is superior to another, as indeed, they are sensitive to different aspects of the data and in some cases can be largely independent. Rather, we emphasize that the concept of a "good" score is nuanced and varies with context." I would like the authors to comment more directly on what should be done. Should this style of analysis be done for every new dataset which can provide a score range that encodes certain relevant variables? Is there some other guideline? It makes sense that there isn't one best choice but the question that starts off the paper doesn't seem to be addressed.
3. The datasets are all electrophysiology datasets whereas comparisons are often also done with fMRI datasets, will these results still hold for these datasets? Especially with the difference in sampling between the methods.

**Questions:**

1. Do the authors have suggestions on how to use these measures? Or do they have a suggestion of what analysis is still needed before picking a measure?
2. Why does ridge regression seem to be independent from Angular Procrustes (Fig 7)?

---

> ### Author Response · Authors · 2024-12-02
>
> Thank you for your positive assessment and helpful comments. We have updated the paper based on your feedback.
>
> Below are several related comments, along with our answers.
>
> > Ridge-Regression seems to be the most widely-used measure in the field although most of the comparisons focus on CKA vs Angular Procrustes. Would be nice to see more commentary on this especially in Figure 7 where it seems independent from angular Procrustes.
>
> In the case of unregularized, uncross-validated linear regression compared to Angular Procrustes (**Figure 7**), our results are consistent with the invariance properties of these metrics. Angular Procrustes is invariant under orthogonal transformations, which is more strict than linear regression, which is invariant under invertible linear transformations. So it is expected that a high Angular Procrustes score implies a high linear regression score but not the other way around. Concerning regularized and cross-validated linear regression, the interpretation of our findings is less straightforward. In general, our method is designed to be widely applicable, i.e. to any differentiable metrics, but doesn’t replace careful theoretical work that studies differences between specific metrics. However, even in cases where there is an apparently straightforward mathematical relationship between canonical versions of two similarity measures, in practice, we find that implementation level details matter and change the allowable ranges of both similarity scores.
>
> For example, our additional results with different variants of representational similarity analysis (RSA), as shown in **appendix C.3**, illustrate that a small difference in the implementation details can make two metrics independent from each other.
>
> > From the start of the paper it seems like it will answer the question: "What metrics should guide the development of more realistic models of the brain?" I would like the authors to comment more directly on what should be done.
>
> We see the opening question from the abstract (referenced in your question) as one of the motivations for the field. However, at this point in time we do not wholeheartedly recommend any of the current similarity measures without some reservations. That being said, not all similarity measures have the same drawbacks and we see angular Procrustes as a reasonable default. But see caveats in, for example, Ostrow et al. NeurIPS 2023 and Khosla & Williams UniReps 2023.
>
> We advise to not just report similarity scores as our findings show that scores by themselves are hard to interpret and their meaning can greatly vary depending on the details of the dataset and the similarity measure. We argue that similarity scores require further interpretation before making any claim based on them. Here we show two concrete ways of doing so, that incorporate the context of the dataset and similarity measure, by optimizing synthetic datasets to determine:
> 1. How strongly task variables are encoded in order to reach the score (**Figure 3**).
> 2. How many PCs need to be captured in order to reach the score (**Figures 4 and 5**).
>
> We have removed the initial opening question from the abstract and updated the paper to hopefully alleviate the confusion about what we are trying to answer. Here are the specific questions that motivated this paper:
> * What constitutes a “good” score? See **Figure 3**.
> * What drives a high similarity score? See **Figures 4, 5** and the analytic results in **section 4.3** as well as **Figure 6**.
> * Does a high similarity score for one measure imply a high score for the others? See **Figure 7**.
>
> > The datasets are all electrophysiology datasets whereas comparisons are often also done with fMRI datasets, will these results still hold for these datasets? Especially with the difference in sampling between the methods.
>
> Since we find that our results on interpreting similarity scores can greatly vary depending on the dataset, we expect that they will be dataset-specific for fMRI data too. That's why we recommend practitioners to apply our analyses on their own datasets, and we provide open source code to facilitate it.
>
> We are interested in seeing our method applied to fMRI, and also calcium imaging datasets as well, in follow-up work. We expect these analyses will also serve to highlight some potentially problematic cross-validation practices in model-data comparisons with fMRI data, for example, whether train-test splits are obtained by randomly shuffling the data or if contiguous blocks are used for testing. In this paper we wanted to show that there exists variability in good scores even across datasets collected using the same method, namely the five electrophysiology datasets that we analyzed.

---

### Official Review · Reviewer_GR1j · 2024-11-04

**Soundness:** 3
**Presentation:** 3
**Contribution:** 3
**Rating:** 5
**Confidence:** 3

**Summary:**

This paper studies several popular methods to quantify the similarity between models and neural data by applying them to five neural data from several studies. The approach is to directly optimize synthetic datasets to maximize their similarity to neural recordings.  The work is of expository nature and there have been several reviews on similarity measures, but this work is model-agnostic and can shed light on how different metrics prioritize various aspects of the data, such as specific principal components or task-relevant information.

**Strengths:**

Similarity measures have played a pivotal role in guiding the development of more realistic models of the brain. This work provides new insights and challenges of such measures.

**Weaknesses:**

This work is of expository nature, so by this nature its advancement in methodology and theory is less significant.

**Questions:**

1. What guidance will you provide to scientists in choosing a suitable similarity measure?

2. How will this work have impact in the way that similarity scores are applied in practice?

---

> ### Author Response · Authors · 2024-12-02
>
> Thank you for your helpful feedback.
>
> > This work is of expository nature, so by this nature its advancement in methodology and theory is less significant.
>
> We noticed that your review primarily mentions the expository nature of our work as a reason for its lower score, but it does not provide further elaboration. To help us better understand and address your evaluation, could you please clarify which aspects of our work influenced your assessment?
>
> > What guidance will you provide to scientists in choosing a suitable similarity measure?
>
> > How will this work have impact in the way that similarity scores are applied in practice?
>
> We have found that CKA can achieve a high score on datasets that fail to encode task relevant information so we suggest that researchers do not rely solely on this commonly used similarity measure when making claims about model-data similarity. In addition, regardless of what similarity measure is used, we encourage researchers to accompany their reported scores with an analysis like the one in **Figure 3** so readers know if the scores are “good” or not, i.e. if it is possible to obtain these scores without encoding task relevant variables. As we’ve demonstrated, a good value for a score isn’t fixed but depends on the similarity measure and dataset so this analysis is something that should be done every time these scores are reported to account for the unique aspects of a particular study.
>
> In addition to this practical guidance for interpreting similarity scores, we also demonstrate a new tool for uncovering what drives high similarity scores. This has exposed shortcomings in some commonly used similarity measures which we see as a necessary first step in improving them! As these new similarity measures are developed it will be important to understand what drives them and our method can play an informative role here.
>
> Given these clarifications, would you consider raising your score for our paper?

---

### Author Response · Authors · 2024-12-02
**General Response**

We thank all the reviewers for the time and effort invested in evaluating our work, and their encouraging comments that find our analyses novel (reviewer c7uz) and fundamental to the field (CM3i). Here we address the comments common to several reviews, and summarize the corresponding changes to the paper. Individual responses to each reviewer are provided separately.

We have updated the abstract to hopefully remove the confusion about our motivations. To clarify, the specific questions that motivated this paper are:
* What constitutes a “good” score? See **Figure 3**.
* What drives a high similarity score? See **Figures 4, 5** and the analytic results in **section 4.3** as well as **Figure 6**.
* Does a high similarity score for one measure imply a high score for the others? See **Figure 7**.

Additionally, we updated the discussion section of the paper to highlight our take-home advice for practitioners, and clarified the text based on specific reviewer feedback.

We advise to not just report similarity scores as our findings show that scores by themselves are hard to interpret and their meaning can greatly vary depending on the details of the dataset and the similarity measure. We argue that similarity scores require further interpretation before making any claim based on them. Here we show two concrete ways of doing so, that incorporate the context of the dataset and similarity measure, by optimizing synthetic datasets to determine:
1. How strongly task variables are encoded in order to reach the score (**Figure 3**)
2. How many PCs need to be captured in order to reach the score (**Figures 4 and 5**)

Overall, we believe these enhancements (including the addition of **new Supplementary Figures S3, S4, and S5**) have significantly improved both the content and presentation of our submission, and it is our opinion that the scientific community would benefit from a timely communication of these results as it is central to a rapidly expanding body of work on representational alignment in connection to both neuroscience and AI.

---

### Meta-Review · Area_Chair_xwwJ · 2024-12-19

**Metareview:**

This paper investigates the properties of a few popular similarity measures and compare them in several neural activity datasets. It addresses the inconsistency across different similarity measures and presents a model-agnostic synthetic dataset optimization to analyze the properties of similarity measures considered. In general, the paper is well written. Most of reviewers agree that optimizing a synthetic dataset to gain insight into how similarity measures behave is novel. The analysis regarding what aspects lead to a high similarity score is important to guide development of new models and to properly apply the modeling results to the brain. On the other hand, there are a few things that should be considered to improve the paper (see reviewers' comments). During the discussion period, there was a strong support to champion this paper.

**Additional Comments On Reviewer Discussion:**

The authors made efforts in responding to reviewers' comments. During the discussion period, there was a strong support to champion this paper.

---

### Decision · Program_Chairs · 2025-01-22

Accept (Poster)